# Systemic Risk in Global Agricultural Markets and Trade Liberalization under Climate Change: Synchronized Crop-Yield Change and Agricultural Price Volatility

**Yoji Kunimitsu [1,*], Gen Sakurai [2] and Toshichika Iizumi [2]** 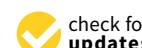

[1]  Institute for Rural Engineering, National Agriculture and Food Research Organization, Tsukuba 305-8609, Japan

[2]  Institute for Agro-Environmental Sciences, National Agriculture and Food Research Organization, Tsukuba 305-8604, Japan; sakuraigen@affrc.go.jp (G.S.); iizumit@affrc.go.jp (T.I.)

*  Correspondence: ykuni@affrc.go.jp; Tel.: +81-29-838-7542

**Abstract:** Climate change will increase simultaneous crop failures or too abundant harvests, creating global synchronized yield change (SYC), and may decrease stability in the portfolio of food supply sources in agricultural trade. This study evaluated the influence of SYC on the global agricultural market and trade liberalization. The analysis employed a global computable general equilibrium model combined with crop models of four major grains (i.e., rice, wheat, maize, and soybeans), based on predictions of five global climate models. Simulation results show that (1) the SYC structure was statistically robust among countries and four crops, and will be enhanced by climate change, (2) such synchronicity increased the agricultural price volatility and lowered social welfare levels more than expected in the random disturbance (non-SYC) case, and (3) trade liberalization benefited both food-importing and exporting regions, but such effects were degraded by SYC. These outcomes were due to synchronicity in crop-yield change and its ranges enhanced by future climate change. Thus, SYC is a cause of systemic risk to food security and must be considered in designing agricultural trade policies and insurance systems.

**Keywords:** agricultural trade liberalization; computable general equilibrium (CGE) model; crop model; food security; simultaneous crop failure; social welfare

## 1. Introduction

Agricultural production is highly influenced by climate conditions [1]; therefore, climate change may add volatility to agricultural production due to crop failures [2,3] or too abundant harvests [4]. Simultaneous crop failures and abundant harvests would enhance synchronized yield change (SYC) for major grains and increase global imbalances in food supply and demand, resulting in extremely volatile agricultural prices [5]. The Intergovernmental Panel on Climate Change (IPCC) suggested that global agricultural prices could increase up to 29% from current levels by 2050, due to the synergy of a decline in agricultural production under future climate change and an increase in the world population [6]. Agricultural price surges should be a risk to food security in the global economy. Thus, clarifying the influence of such risk is an important academic and agricultural policy issue.

Wright [7] analyzed the causes of past spikes in agricultural commodity prices and showed that speculation and rising oil prices were not reasons behind price spikes. The actual explanation was the imbalance in supply and demand, in addition to changes in the grain stock level of global markets. Meanwhile, Headey and Fan [8] stated that the rise in agricultural prices in 2007 was strongly

influenced by factors other than the supply–demand balance in the food market, and that the impact of supply shocks caused by climate change was relatively small. However, they could not ignore climate shocks as a causative factor. Based on these previous findings, it is useful to apply analytical methods that consider supply and demand in the food market in any analysis of agricultural price volatility.

The computable general equilibrium (CGE) model is a powerful analytical tool for analyzing the supply–demand equilibrium and equilibrium prices simultaneously. Many previous studies in the field of agricultural and environmental economics have applied CGE models to assess agricultural trade policy [9], earthquake disasters [10,11], climate change [4,12–15], and environmental policy [16–19]. Notably, in the field of environmental evaluation, some previous studies employed a method that combined global climate model (GCM), crop model, and CGE model to evaluate climate impacts on economies [12,15,19]. However, there seem to be a few quantitative studies that applied CGE model to analyze the systemic risks in global food markets under future climate change.

Nevertheless, Tanaka and Hosoe [20] and Hosoe [21] used a global CGE model to examine the impact of productivity shocks and the effects of trade liberalization on global food markets by performing a Monte Carlo simulation analysis. Their results showed that food-importing countries further increased imports after trade liberalization, but the decline in agricultural prices raised the social welfare level and eliminated negative influences of domestic production decline. Therefore, these studies concluded that trade liberalization could decrease domestic agricultural price volatility despite overseas productivity changes and would not reduce the food security level of importing countries such as Japan.

The conclusions of the research by Tanaka and Hosoe [20] and Hosoe [21] are consistent with portfolio theory in that overall fluctuation can be reduced by diversifying the combination of products from different areas with different price fluctuation patterns, regardless of import and domestic production. However, these studies have not analyzed the impact of correlated shocks among major grains or with other countries. Theoretically, shocks correlated across traded products increase fragility in the market and can be a "systemic risk" [22]. In other words, risk hedging to mitigate future price fluctuations can be accomplished by combining stocks whose prices fluctuate independently. However, the combined stock price may drastically fall if there is a correlation in the price fluctuations. Systemic risk originates in tightly coupled systems and is characterized by interlocking effects, tipping points and nonlinear developments [23]. If climate change enhances a correlation among crops or among producing countries, agricultural trade liberalization may not be useful as a mechanism to hedge risk under climate change.

This study analyzes whether future climate change increases systemic risk via SYC. When and if such systemic risk exists, this study attempts to evaluate quantitively the influence of SYC (a source of systemic risk) on the global food market with consideration of trade liberalization. The methodological features of this study are as follows. First, we integrate the CGE model and two crop models to treat prediction results of five GCMs for a more general prediction-based assessment. Second, economic impacts of climate shocks generated by these models are compared to the results from the stochastic simulation method to quantify the difference between SYC and non-SYC situations. Policies for addressing systemic risks may include the allocation of production sources, insurance systems to mitigate crop failures, and government involvement to improve forecast accuracy.

The remainder of this paper is organized as follows. Section 2 explains the methods used in this analysis, including the crop model and the CGE model. Section 3 demonstrates the chronological robustness of the SYC structure in the global production systems of four major grains and detects the impacts of systemic risk on food markets caused by SYC, considering future climate change and agricultural trade liberalization. Based on these results, Section 4 notes some policy implications. Finally, Section 5 summarizes the results of the analysis and presents the conclusions.

## 2. Materials and Methods

The analysis measures the degree of SYC among countries and among crops under future climate change using the predictive results of the crop model, and it verifies the chronological robustness of SYC. Subsequently, fluctuations in agricultural prices are calculated by inputting the results of the crop model into a global CGE model. These economic results are then compared with a no-correlation case, in which SYC among countries and crops does not exist, by applying a Monte Carlo simulation analysis according to Hosoe [21]. Figure 1 shows these analytical procedures.

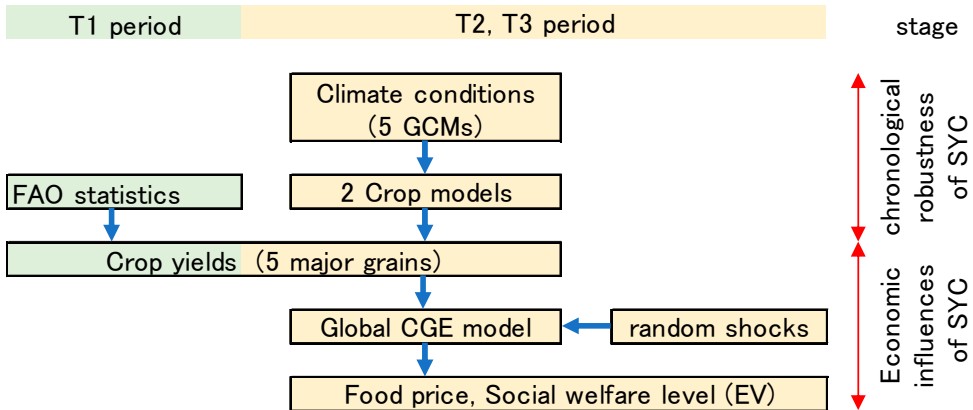

**Figure 1.** Analytical framework. T1 period is 1961–2014, and T2 and T3 periods are, respectively, 2015–2050 and 2051–2100. FAO statistics refer to the crop-yield statistics of the Food and Agriculture Organization (FAO). GCM is global climate model, and CGE model is computable general equilibrium model. EV means equivalent variation.

### 2.1. Crop Model and Data

The possibility of simultaneous crop failure has already been verified by previous studies using historical data [3], as well as future prediction data [2]. This study instead examines the chronological robustness of correlation coefficients on annual yield change, including small fluctuations in addition to crop failure and too abundant harvests.

Crop-yield data of four crops—rice, wheat, maize, and soybeans—were prepared over three periods: T1 (1961–2014), T2 (2015–2050), and T3 (2051–2100). T1 corresponds to the period covered by the crop-yield statistics of the Food and Agriculture Organization (FAO); FAO data were detrended to remove technological progress (Appendix A). T2 and T3 are yield periods under future climate change; yield data for these periods were produced by crop models of four major crops. The crop models employed in this study are PRYSBI2 [24] and pDSSAT [25,26]. The PRYSBI2 is a hybrid type of process-based and the empirical model, consisting of biological equations with observed parameters from field experiments, as well as uncertain parameters estimated using the Markov chain Monte Carlo method with statistical yield data. Meanwhile, pDSSAT is a pure process-based model that replicates crop growth stages based on biological functions. These process-based models can estimate yields by considering the changes in daily climate conditions predicted by GCMs. Both models were employed in the Agricultural Model Intercomparison and Improvement Project (AgMIP), and the yield data of these crop models were estimated according to the AgMIP common protocol [25]. Müller et al. [27] showed that the reproducibility of PRYSBI2 was 0.260 n.s., 0.303 n.s., 0.527 **, and 0.279 n.s for maize, wheat, rice, and soybean, respectively, whereas that of pDSSAT was 0.888 ***, 0.652 ***, 0.215 n.s., and 0.496 ** (in the same order). Here, ***, **, and n.s. indicate significant at $p < 0.001$, significant at $p < 0.05$, and not significant at $p < 0.1$. These numbers were calculated from time series of crop-yield estimations and FAO's statistical data in the global scale after detrending. Some crops had insignificant correlations due to small sample size, but generally temperature and precipitation can only explain approximately 30% of year-to-year variations in the average global yields of measured crops [1], which

corresponds to a correlation coefficient of 0.55. Considering this value, the reproducibility of the two crop models meets the level available for analysis.

In the estimation, daily climate conditions and $CO_2$ fertilizer effects were considered according to the representative carbon pathway (RCP) scenario, RCP8.5, that corresponds to the highest $CO_2$ concentration among scenarios mentioned in the IPCC report. Although there are many objections in the field of crop science regarding the magnitude of the effects of $CO_2$ fertilizer [28], such effects can ease the future degradation risk of crop yield and avoid overestimating risk; therefore, this study considered the effects of $CO_2$ fertilizer. The technology level was fixed at its 2000 level, which was before the start year of the simulation, to eliminate the effects of afterward technological progress, such as breed improvement. Influences of flood damage caused by heavy precipitation were ignored, although the effects of drought were considered through changes in soil moisture caused by rainfall.

Daily climatic conditions forecasted by the five GCMs (i.e., HadGEM2-ES, IPSL-5 CM5A-LR, MIROC-ESM-CHEM, GFDL-ESM2M, and NorESM1-M) were entered into the two crop models for the annual crop-yield forecast of each country from 2007 to 2099. Five GCMs' data on the RCP 8.5 scenario were obtained from the Coupled Model Intercomparison Project 5 (CMIP5) [29], and these were bias corrected according to Hempel et al. [30].

The analysis, then, calculated the correlation coefficients of yields in T1, T2, and T3 periods. As it was useless to compare the climate data of each GCM in a specific year, the correlation coefficient was calculated from the sample corresponding to each crop model and each GCM (2 crop models × 5 GCMs), consisting of year periods in T1, T2, and T3. Therefore, the sample size was the total number of 2 models × 5 models × period, excluding the year when the crop models' estimation was not successful.

Thereafter, we selected statistically significant combinations of regions and/or crops for which the correlation coefficients of annual yields are significant in each period and investigated whether these combinations in one period can continue to the next period. The number of combinations, which retain significance in correlation coefficients between two periods, was thought to show the degree of SYC robustness.

The analyzed area is a total of 38 countries and/or regions, including 29 countries that are major producers and importers of the four target grains and 9 regions that are integrated based on geographical proximity (Table 1). These integrations are due to a limitation in the ability of CGE model explained later. For simplicity, each country and/or region is simply referred to as a "region" hereafter.

**Table 1.** Aggregated regions for analysis.

| No. | Identifier | Country or Region | No. | Identifier | Country or Region |
|---|---|---|---|---|---|
| 1 | AUS | Australia | 20 | URY | Uruguay |
| 2 | CHN | China | 21 | XSM | Rest of South America |
| 3 | JPN | Japan | 22 | XCA | Rest of Central America |
| 4 | KOR | Korea Republic of | 23 | FRA | France |
| 5 | IDN | Indonesia | 24 | DEU | Germany |
| 6 | PHL | Philippines | 25 | GBR | United Kingdom |
| 7 | THA | Thailand | 26 | XEF | Rest of Western Europe |
| 8 | VNM | Vietnam | 27 | ROU | Romania |
| 9 | BGD | Bangladesh | 28 | RUS | Russian Federation |
| 10 | IND | India | 29 | UKR | Ukraine |
| 11 | PAK | Pakistan | 30 | XER | Rest of Europe |
| 12 | XAS | Rest of ASIA | 31 | IRN | Islamic Republic of Iran |
| 13 | CAN | Canada | 32 | TUR | Turkey |
| 14 | USA | United States of America | 33 | XWS | Rest of Middle East |
| 15 | MEX | Mexico | 34 | EGY | Egypt |
| 16 | ARG | Argentina | 35 | XAC | South Central Africa |
| 17 | BOL | Bolivia | 36 | XEC | Rest of Eastern Africa |
| 18 | BRA | Brazil | 37 | ZAF | South Africa |
| 19 | PRY | Paraguay | 38 | XTW | Rest of the World |

## 2.2. CGE Model for Estimation of Agricultural Price Volatility

Annual agricultural price is estimated by a static global CGE model [31] based on the Global Trade Analysis Project (GTAP) 9 database [32], considering the supply–demand balance in the global crop trade markets. The detailed structure of the model is explained by Lanz and Rutherford [31]; hence, this study notes only the following points as the model's primary features.

First, production is formulated by a nested constant-elasticity-of-substitution (CES) function. Value added is produced by four input factors (i.e., labor, capital, farmland, and natural resources), and total gross production is calculated by combining value added with intermediate inputs, comprised of import and domestic goods. Domestic production and export are further divided from total gross production based on the constant-elasticity-of-transformation (CET) function. Imports are inserted into the production process by a CES-type function based on Armington's assumption [33]. Second, consumption is formulated by a linear expenditure system (LES)-type function that treats basic consumption and variable consumption separately. Variable consumption is determined by considering the substitutability of each consumption good that is produced from domestically produced and imported goods according to Armington's assumption. Similarly, investment and government consumption are defined by a Leontief-type function that combines consumption goods comprised of domestic and imported goods. Third, production tax, production factors tax, intermediate input tax, consumption tax, public sector purchase tax, investment tax, export subsidy, and import tariff are all considered to cover the tax systems of each country in the world.

Original GTAP industrial sectors are combined into 12 sectors (Table 2), and countries in the world are merged into 38 regions (Table 1). The parameters of each function are calibrated by the data of 2011 in the GTAP 9 database. The substitution elasticities for production, consumption, government consumption, and trade are also derived from the GTAP 9 database, as well as the Frisch parameter in consumption.

**Table 2.** Aggregated industrial sectors for simulation analysis.

| No | Identifier | Industrial Sectors | No | Identifier | Industrial Sectors |
|----|-----------|---------------------|----|-----------|---------------------|
| 1 | PDR | Paddy rice | 7 | MIN | Forestry, fishery, and mining |
| 2 | WHT | Wheat | 8 | VOL | Vegetable oils and fats |
| 3 | GRO | Other cereal grains (including maize) | 9 | PCR | Processed rice |
| 4 | OCR | Other crops | 10 | OFD | Other food products |
| 5 | OSD | Oil seeds (including soybeans) | 11 | MAN | Manufacturing |
| 6 | OAP | Animal products and other agriculture | 12 | SEV | Service |

## 2.3. Simulation Method

The global CGE model is exogenously subjected to disturbances caused by the yield changes of four crops. We assume that these disturbances affect the production of each crop through effective farmland productivity (EFP) in the CES-type cost function (Appendix B). The disturbances here are seemingly time-series data corresponding to the yearly data produced randomly or predicted by crop models based on GCMs' prediction. However, the shock of the disturbances is assumed to converge in 1 year; therefore, the simulation performs repeatedly static analyses according to the number of disturbance data. The simulation cases considered are as follows.

Case 1 (Random and regionally independent disturbances, non-SYC): The yield changes create random and regionally independent disturbances in EFP, and the current trade structure is maintained (no change in import tariffs and export subsidies). One thousand random shocks are generated based on a lognormal distribution as an assumption. The standard deviation given at the time of random number generation is set to the standard deviation of the estimated yield by the crop models for the period 2007–2014 to match the initial disturbance range with other cases. This case is an estimate during the T2 and T3 periods without SYC and is used as reference for the subsequent Cases 2–6.

Case 2 (SYC during T2): This case considers the SYC of the four crops that would be expected during the T2 period under climate change. Crop yields with SYC are then assumed to change EFP as follows:

$$EFP_{i,r,t} = \left( YE_{i,r,t} / \overline{YE}_{i,r} \right), \tag{1}$$

where, *i*, *r*, and *t* represent the four crop categories, countries, and year, respectively. Further, *YE* is the yield estimated by the crop model, $\overline{YE}$ is an average of *YE* during the years 2007–2014 and is used as the referenced level in the simulation. As there are 36-year estimations by 2 crop models with 5 GCMs, the total number of iterations is 353 (=36 years × 2 crop models × 5 GCMs' inputs −7 as unobservable data) in the simulation. Furthermore, the current trade structure is maintained.

Case 3 (SYC during T3): Yield data estimated by crop models are during T3, and all other settings are the same as in Case 2. The total number of iterations is 467 (49 years × 2 crop models × 5 GCMs' inputs −23 as unobservable data).

Cases 4, 5, and 6 (Agricultural trade liberalization cases): These cases correspond to agricultural trade liberalization in Cases 1, 2, and 3, respectively. All regions' import tariffs and export subsidies for agriculture and food sectors (PDR, WHT, GRO, OCR, OSD, OAP, OFE, VOL, and PCR) are set to 0. Other settings are the same as in Cases 1, 2, and 3, respectively.

Table 3 summarizes the setting conditions for each described simulation case. From the difference between any two cases in this table, the effects of SYC or trade liberalization can be calculated when another condition is set equal. For example, the difference between Case 3 and Case 1 shows the effect of strong SYC when trade liberalization is not considered; meanwhile, the difference between Case 6 and Case 3 shows the effect of trade liberalization with strong SYC. Hereinafter, the notation "Cases $X_1/X_2$" implies the ratio of Case $X_1$ against Case $X_2$, whereas the notation "Cases $X_1 - X_2$" indicates difference between Cases $X_1$ and $X_2$.

**Table 3.** Summary of simulation conditions.

| Simulation Conditions | Case 1 | Case 2 | Case 3 | Case 4 | Case 5 | Case 6 |
|---|---|---|---|---|---|---|
| Synchronized yield change | None | Weak | Strong | None | Weak | Strong |
| Trade liberalization | None | None | None | Adopted | Adopted | Adopted |

## 3. Results

### 3.1. Robustness of the SYC Structure

Table 4 presents the frequency of the correlation coefficients between regions by magnitudes and periods, calculated using available data from regions on annual crop-yields. The percentage values are the ratios of regions' number classified according to the magnitude of the correlation coefficient against the total combination number, nt. Similarly, the values presented in Table 5 are calculated based on correlation coefficients between crops.

As shown in Table 4, correlation coefficients were statistically significant in many combinations of regions, showing a high occurrence possibility of similar climatic conditions between two regions, i.e., SYC among regions was recognized. Correlation coefficients between crops (Table 5) also show the existence of SYC between crops. Furthermore, there were more combinations of regions or crops with a positive correlation than a negative correlation. This happened due to the following two influences. First, when climate change progressed worldwide, yields in each region simultaneously decreased or increased, creating the chronological similar trend of yield change in each region where the same crop was planted. Second, in addition to an increase in fluctuation of climate conditions in many regions estimated by GCMs, crop-yield changes became more sensitive to changes in climate conditions. When climatic conditions were close to the biological threshold level, even a small change in temperature caused growth disorders and increased yield variability. Such tendency led to simultaneous crop failures in many regions. Hence, the degree of SYC was increased with the progress of global warming.

**Table 4.** Frequency in magnitude of correlation coefficient (*r*) between regions by crops and period.

| Crops | Periods | *n/nt* | r < −0.4 | −0.4 < r < −r_NZ | −r_NZ < r < r_NZ | r_NZ < r < 0.4 | 0.4 < r |
|---|---|---|---|---|---|---|---|
| Rice | T1 | 35/595 | 16.6% | 4.0% | 54.5% | 4.5% | 20.3% |
| | T2 | 35/595 | 0.3% | 5.2% | 40.0% | 21.5% | 32.9% |
| | T3 | 35/595 | 11.3% | 7.7% | 13.3% | 23.2% | 44.5% |
| Wheat | T1 | 35/595 | 4.9% | 1.7% | 82.8% | 4.2% | 6.4% |
| | T2 | 35/595 | 4.9% | 19.2% | 35.6% | 30.3% | 10.1% |
| | T3 | 35/595 | 15.3% | 9.4% | 11.3% | 15.6% | 48.4% |
| Maize | T1 | 37/666 | 9.5% | 3.5% | 55.1% | 3.3% | 28.7% |
| | T2 | 36/630 | 0.0% | 6.7% | 59.7% | 30.3% | 3.3% |
| | T3 | 36/630 | 0.2% | 4.6% | 9.0% | 30.0% | 56.2% |
| Soybeans | T1 | 34/561 | 4.5% | 2.1% | 86.1% | 2.5% | 4.8% |
| | T2 | 36/630 | 0.0% | 7.3% | 31.4% | 34.8% | 26.5% |
| | T3 | 36/630 | 6.5% | 7.8% | 16.0% | 24.6% | 45.1% |

Note: The percentage value is a ratio between the number of correlation coefficients falling within the range and the total number of calculations. Further, *n* is the total number of regions where data were obtained, *nt* is total number of combinations that are calculated by $n \cdot (n − 1)/2$, and r_NZ shows the magnitude of correlation coefficients that are significantly different from 0 as compared to the t-statistic value at a 1% significance level. The sample size for calculation in each period is T1: 54 (1961–2014), T2: 356 = 36 (2015–2050) × 5 (GCMs) × 2 (crop models) −4 (unobserved data), and T3: 468 = 49 (2051–2099) × 5 (GCMs) × 2 (crop models) −22 (unobserved data).

**Table 5.** Frequency in magnitude of correlation coefficient (r) between crops by period.

| Periods | *n/nt* | −0.4 < r < −r_NZ | −r_NZ < r < r_NZ | r_NZ < r < 0.4 | 0.4 < r |
|---|---|---|---|---|---|
| T1 | 4/6 | 0.0% | 50.0% | 50.0% | 0.0% |
| T2 | 4/6 | 0.0% | 0.0% | 83.3% | 16.7% |
| T3 | 4/6 | 33.3% | 0.0% | 50.0% | 16.7% |

Note: *n* is the total number of crops, and the sample size for the calculation in each period is T1: 972 = 54 (years in 1961–2014) × 18 (regions), T2: 9720 = 36 (years in 2015–2050) × 5 (GCMs) × 2 (crop models) × 27 (regions), T3: 12,000 = 50 (years in 2051–2100) × 5 (GCMs) × 2 (crop models) × 24 (regions). Other notations are the same as in Table 4.

The stability of the crop yield's correlation structure was verified by selecting the statistically significant correlation coefficients from each period. We counted combination numbers where the correlation coefficients between T1 and T2 or between T2 and T3 were statistically nonzero at the 1% probability level ("rr_NZ") and positive ("rr_+") in both periods. Then, we calculated the ratio of these combination numbers against the total number of combinations (*nt*). Among these, combinations with higher correlation coefficients in the later period than in the earlier period ("rr_1") indicate that SYC globally became stronger.

Table 6 summarizes the robustness of the correlation coefficients among regions between T1 and T2 and between T2 and T3. In both T1–T2 and T2–T3, the ratios of "rr_NZ" were obvious, and specifically, the ratios of T2–T3 in analyzing the four crops cross sectionally were 37.5–60.7%. The ratios of "rr_+" in T2–T3 were higher than in T1–T2 for all crops, showing that synchronicity in yield change became stronger over time. Actually, T1 was based on observed crop yield and included short time disturbances that could not be eliminated by detrending, such as rising oil prices [34] and effects of conflicts [7]. Therefore, the correlation in T1 was weaker than in T2 or T3 in which yield changes were influenced only by climate conditions. In T2–T3, rice and soybean had a higher percentage of combinations in "rr_+," as well as percentages of "rr_1" than for the other two crops.

Tigchelaar and Battisti [2] found that the likelihood of simultaneous crop failure in maize production increased chronologically due to future climate change. Their findings are consistent with our analysis, though only about maize, as shown by the high correlation coefficients in Table 4 and the chronological change (ratios of rr_1) in Table 6.

Table 7 shows the robustness of the correlation coefficients among crops in the same way as Table 6. Here, rr_NZ and rr_+ were higher in T2–T3 than in T1–T2, showing a similarity with the inter-regional tendency. From the results on inter-regional and intercrop correlations, our analysis demonstrates that the yields of four crops in each region of the world tend to fluctuate in the same

direction. In other words, the SYC structure is robust not only in the past period but also in the future under climate change.

**Table 6.** Degree of corresponding regions with positive correlations between two periods.

| Crops | Periods | *nt* | rr_NZ | rr_+ | rr_1 |
|---|---|---|---|---|---|
| Rice | T1→T2 | 595 | 30.9% | 16.8% | 5.9% |
| | T2→T3 | 595 | 55.8% | 50.1% | 41.5% |
| Wheat | T1→T2 | 595 | 12.6% | 6.2% | 2.5% |
| | T2→T3 | 595 | 60.7% | 38.3% | 37.3% |
| Maize | T1→T2 | 630 | 14.6% | 10.3% | 0.8% |
| | T2→T3 | 630 | 37.5% | 33.5% | 32.9% |
| Soybeans | T1→T2 | 561 | 6.4% | 3.4% | 1.6% |
| | T2→T3 | 630 | 50.0% | 46.5% | 42.7% |

Note: "*nt*" is the total number of combinations, and "rr_NZ," "rr_+," and "rr_1" show which correlation coefficients are statistically nonzero in both periods at the 1% probability level, which are positive in both periods, and which are bigger in latter periods than in former periods, respectively.

**Table 7.** Degree of corresponding crops with positive correlations between two periods.

| Periods | *nt* | rr_NZ | rr_+ | rr_1 |
|---|---|---|---|---|
| T1→T2 | 6 | 50.0% | 50.0% | 50.0% |
| T2→T3 | 6 | 100% | 66.7% | 33.3% |

Note: The total number of combinations, *nt*, was 6 (=3 × 4/2). Other notations are the same as in Table 6.

*3.2. Initial Shocks in EFP Given to the CGE Model*

Figure 2 shows the average and the standard deviation of EFP. FAO_T1 is the detrended FAO's yield data, and Cases 1 and 4 are the random disturbance cases without SYC (non-SYC). Cases 2 and 5 and Cases 3 and 6 are SYC cases for the T2 and T3 periods, respectively. In the simulation, the same EFP was set for the case numbers connected by commas (i.e., Cases 1 and 4, Cases 2 and 5, and Cases 3 and 6). Results from the entire world average and only five major regions (i.e., the United States, China, Brazil, Japan, and France) are presented in Figure 2 due to space reasons. These regions were major producers and major importers in the world for four crops targeted in this study.

The EFP values of all 38 regions and those estimated by each crop model (PRYSBI2 and pDSSAT) are shown in the Supplementary Materials (Figures S1–S8). By comparing the results of two crop models, although the ranges of EFP by PRYSBI2 were bigger than those by pDSSAT, similar tendencies of EFP change were found in two models on the four crops. Therefore, we used the average value, standard deviation, calculated by averaging the two models' estimations, and maximum or minimum values, which were the maximum or minimum values of each model's estimations as the variable of interest. When calculating these indices, we were, of course, careful not to mix data from different crop models, as well as different GCMs, and to treat data from each model separately.

In this figure, FAO_T1 and Cases 1 and 4 marked almost the same level in average EFP, although these two cases were not reproductions of T1 represented by FAO's actual data. The coincidence in average values between Cases 1 and 4 and FAO_T1 indicates that these cases did not deviate significantly from the past actual situations and were reasonable predicted values. The standard deviations in the two cases were different in some regions, but on a world average, the ratio of the standard deviations between each case and FAO_T1 was within the range of 0.6–1.6. In the simulation, the non-SYC disturbances were generated based on the values of the crop model from 2007 to 2014, not based on FAO's actual data, and consequently, differences from FAO_T1 did not affect the subsequent simulation results.

By comparing the non-SYC case of Cases 1 and 4 with the SYC case of Cases 2 and 5 and Cases 3 and 6, the average EFP of rice (PDR) and soybean (OSD) were higher in the SYC case, but that of

wheat (WHT) and maize (GRO) were lower than in the non-SYC case. In particular, these tendencies were remarkable in Cases 3 and 6 due to the differences in the reflection characteristics of each crop to climatic conditions.

The fluctuation range shown by the standard deviation of EFP became larger in the SYC cases, and especially, Cases 3 and 6 marked the widest fluctuation range. As the average world temperature continues to rise toward the T3 period, which corresponds to Cases 3 and 6, the above tendency implies that future climate change will widen the fluctuation range of crop yield. Among crops, the fluctuation range of OSD was the largest, while that of GRO was small compared to other crops, due to the differences in climatic characteristics of each crop.

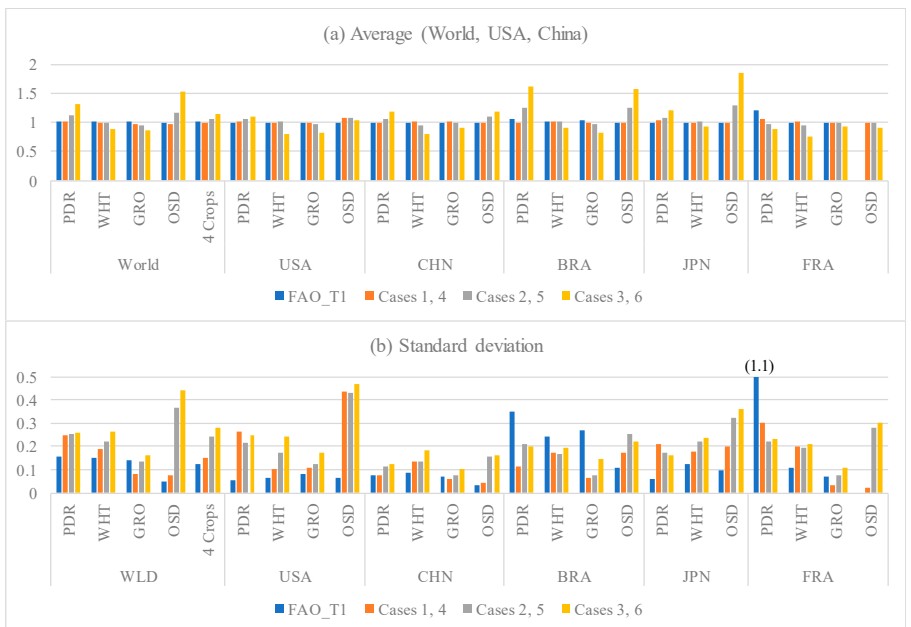

**Figure 2.** Average and standard deviation of effective farmland productivity (EFP) by crops. "FAO_T1" refers to detrended actual yield of the Food and Agriculture Organization (FAO) statistics during the T1 period. Cases 1 and 4 correspond to random disturbances (non-SYC); Cases 2 and 5 and Cases 3 and 6 correspond to synchronized yield change (SYC) produced by the crop models. The values of the whole world (WLD) are the entire world average of each region's values, and values of "4 Crops" are the average of four crops' EFPs. The standard deviation of the France's rice (PDR) is by far the largest, and the value is shown between parentheses above the bar graph.

### 3.3. Influence of SYC on Agricultural Price

Figure 3 compares the (a) average level, (b) standard deviation, and (c) highest level of estimated agricultural price (P_Agr) for Cases 1, 2, and 3. In this figure, according to the settings of CGE model, the price is represented by an index with 2011 as 1.0.

By comparing the non-SYC case (Case 1) with the SYC cases in T2 period (Case 2) and the T3 period (Case 3) in Figure 3, the average, standard deviation, and highest price levels show the similar tendency. Cases 1 and 2 remained at almost the same level, but Case 3 significantly became highest. In other words, if future global warming progresses within 2 °C as in the T2 period, price and its fluctuation will moderately increase. However, when the average temperature reaches to the high level of 4 °C or higher in the T3 period, the price will rise sharply, and the instability of price fluctuation will significantly increase.

Comparing the differences in these indices, the standard deviation and the highest price in Figure 3 marked large difference between cases than the average price. For example, in the United States, the average price in Case 3 was 1.17 times higher than that in Case 1, while its standard deviation was 10 times larger, and its highest price was 2.5 times higher than in Case 1. The average price is related to

the average yield, while the standard deviation and highest price are related to the fluctuation range of the yield. Therefore, it can be said that the factors that result in price instability are the expansion of the fluctuation range of yields and the synchronicity of fluctuations under global warming, not the level of average crop yields.

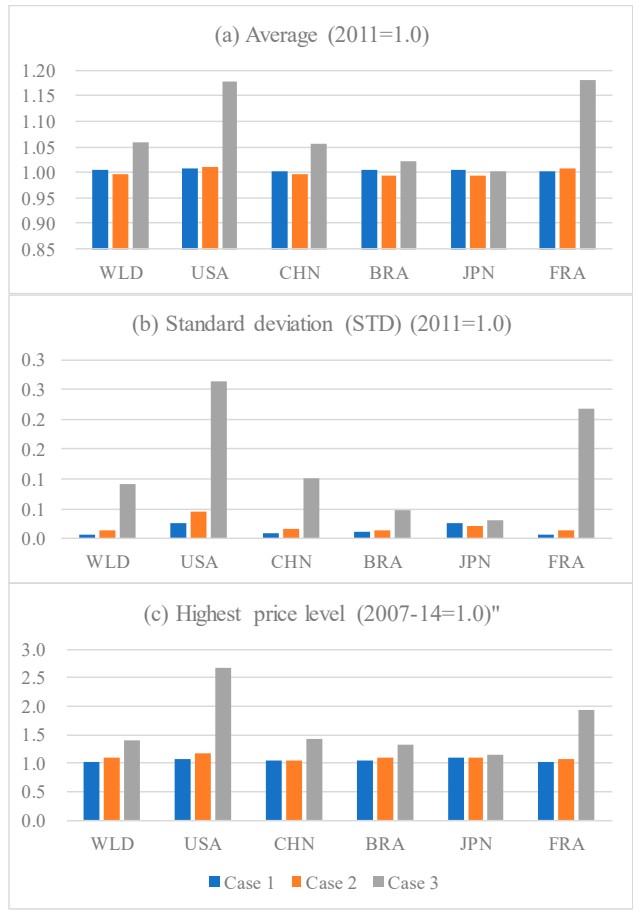

**Figure 3.** Influence of SYC on the price of agriculture products (P_agr). "WLD" shows the world weighted average of the domestic agricultural price by the production value of each country. Similar comparison by all regions and two crop models is shown in the Supplementary Materials (Figures S9 and S10).

*3.4. Effects of Agricultural Trade Liberalization on Agricultural Price Volatility*

Figure 4 shows the changes in agricultural price (P_agr) in major countries before and after trade liberalization. To illustrate the net impact of trade liberalization, this figure focused on the ratios of two cases, such as Cases 4 and 1 (Cases 4/1), Cases 5 and 2 (Cases 5/2), and Cases 6 and 3 (Cases 6/3). Cases 4/1, Cases 5/2, and Cases 6/3, respectively, show the net effect of trade liberalization under non-SYC, weak SYC in T2 period, and strong SYC in T3 period.

Although the values are only shown in the Supplementary Materials (Figure S11) due to space considerations, net exports (i.e., exports minus imports) of the agriculture and food products increased in most food-exporting regions and decreased in food-importing regions, and agricultural trade expanded after trade liberalization. Hence, an increase in imports caused a reduction in domestic production in most food-importing countries.

Based on these results, the following can be observed in Figure 4. As shown by Cases 4/1 in the case of non-SYC, trade liberalization increased benefits of food-importing regions. For example, in Japan, as one major importers, agricultural price was decreased by 10% (1–0.90) after trade liberalization,

and the range of price fluctuations was narrowed by 97% (1–0.03). As a decrease in standard deviation indicates stabilization of price volatility, the above influences are positive effects for Japan's consumers.

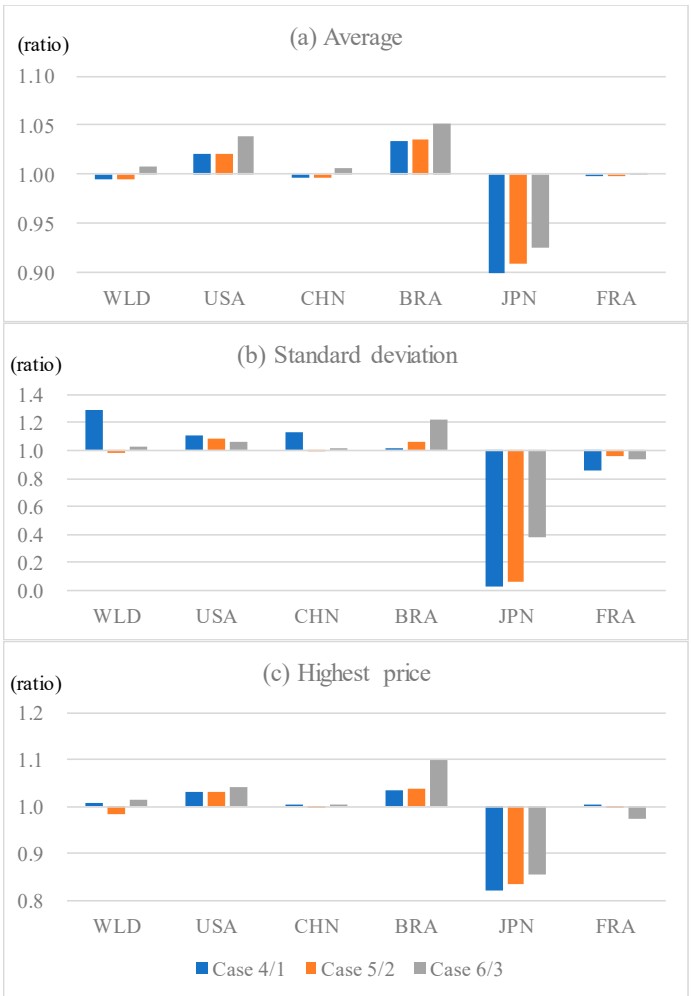

**Figure 4.** Net effects of trade liberalization in each situation (non-SYC, SYC in T2, and SYC in T3). Cases $X_1/X_2$ means ratio of Case $X_1$ against Case $X_2$. Net effects of trade liberalization comparing by regions and two crop models are shown in the Supplementary Materials (Figures S12 and S13).

Meanwhile, in food-exporting regions, such as the United States and Brazil, agricultural price rose after trade liberalization, and the fluctuation range of the price increased. As the export volume increased due to trade liberalization, these agricultural exporters were more affected by the high price traded in importing countries. In China and the whole world (WLD), the prices of agricultural products after trade liberalization were almost the same as those before liberalization, and the price fluctuation range was slightly increased. This is because the effects of liberalization in export and import regions were offset in WLD, and those of liberalization in export and import agricultural products were offset in China that exports maize but imports wheat and soybean. In France, the average price change was slight, similar to China, but the standard deviation decreased after liberalization. Even in France, a similar offset between import and export products was present, but its degree was different from China's.

The comparison between Cases 4/1, 5/2, and 6/3 shows that the decrease degree in average price in Japan became lower, indicating that the ratio of two cases changed from 0.9 (10% decrease) to 0.91 or 0.92 (8% decrease). The ratio of two cases in the standard deviation of price fluctuations also changed from 0.03 to 0.06 or 0.38, and the decrease degree in the fluctuation range by trade liberalization became lower in the SYC case than the non-SYC case. Note that a ratio close to 1 means that little change

occurred in the fluctuation range of the latter case and a small ratio indicates that the fluctuation range was more significantly narrowed in the latter case. Thus, the positive effect of Japan's trade liberalization in the SYC case became smaller than in the non-SYC case.

Furthermore, in terms of export regions, such as the United States and Brazil, the ratios of two cases in average price and highest price were larger in the SYC case than those in the non-SYC case. Specifically, in Brazil, the ratio even in the standard deviation also became larger in the SYC case than in the non-SYC, showing an expansion in the annual fluctuation range. This means that the negative effects of trade liberalization were exacerbated due to SYC. In the United States, however, the annual fluctuation range decreased slightly by SYC, indicating a mitigation of the negative effect of liberalization. Similarly, in WLD and China, the effects of liberalization shown by average price and highest price were exacerbated, but the liberalization effects shown by the annual fluctuation range were improved by SYC.

Overall, for domestic consumers, trade liberalization has a positive impact on food-importing regions, making price levels lower and price fluctuation more stable. Conversely, for food-exporting regions, trade liberalization causes a negative impact due to increasing domestic prices. Meanwhile, SYC reduces the positive effects in importing regions and exacerbate the negative effects of rising prices in exporting regions. Although liberalization affects oppositely in import and export regions, SYC has a negative impact on the effects of liberalization in both import and export regions.

### 3.5. Influences of SYC on Social Welfare Levels

To show the macroeconomic influence of SYC, we examined social welfare level measured by equivalent variation (EV), in accordance with Hosoe [21]. EV in each case is the change in the monetary value of utility level from the starting year level, corresponding to the calibration year of 2011. As the average of disturbances in Cases 1 and 4 was set to 1, which was the same as the calibration year, the average EV for Cases 1 and 4 became approximately 0 and can be regarded as unchanged from 2011, although variations in EV do exist throughout the years.

Figure 5 shows the average, minimum, and standard deviation of EV. Since EVs in non-SYC of Case 1 were almost 0 and show no change in the social welfare level from the present level, EVs of Cases 3, 4, 5, and 6 were subtracted from or divided by the EV of Case 1 to measure the changes from non-SYC without trade liberalization.

Case 3 minus 1 shows that social welfare levels declined in most countries due to SYC in T3 period. In the worst case shown by Figure 4b, social welfare levels declined by US$62 billion in the United States, US$37 billion in China, and more than US$160 billion in WLD. Conversely, Case 4 minus 1 shows that trade liberalization under non-SYC increased the social welfare level in many regions. In particular, there were remarkable increases in Japan, where agricultural tariff rates were high, and in the United States and Brazil, which export foods. In food-exporting countries, although domestic agricultural price increased (Figure 3) and consumers' surplus decreased, household income could increase due to an expansion of export, and then, EV increased due to income effects.

Considering both climate change and trade liberalization (Case 5 or Case 6 minus Case 1), the average EV increased in the United States, Brazil, and Japan, similar to the difference between Cases 4 and 1. In WLD and food-importing regions such as Japan, agricultural price was decreased by trade liberalization, and liberalization could overwhelm the negative effects of SYC under climate change. However, these positive effects of trade liberalization were decreased by SYC in T3 by 12% (US$16.5 billion/US$18.8 billion) in Japan. Furthermore, at minimum EV (Figure 5b), the EVs of the United States, China, and WLD in Case 6 were worse than those in Case 1 (and also in the initial level of the simulation) because of an increase in the fluctuation ranges of EV under SYC.

The EV's standard deviation indicates that the variations in Cases 3 and 6 were much larger than in Case 1, and those in Cases 4 and 5 were slightly larger than in Case 1. This implies that both extreme climate change and trade liberalization widened the gap between a good and a bad year of social welfare level in all regions. These results differ slightly from the results of previous

study [21], which demonstrated a reduction in deviations in Japan's social welfare level by trade liberalization. Since trade liberalization for food-importing region could pool the risk origins of crop-yield change occurring in different countries, the volatility of agricultural price was reduced by agricultural trade liberalization, showing the same tendency as the previous study. However, such effects were weakened in EV due to different trade liberalization schemes. Liberalization among all regions was assumed in this study and marked weaker diversification effects of importing regions than the unilateral liberalization scheme set in the previous study. This happens due to competition with other regions for imports under fully liberalization scheme. Furthermore, such effects of trade liberalization in the EV's fluctuation range were reduced by the decrease in price effects caused by SYC under climate change. Such influences have not been evaluated by previous studies.

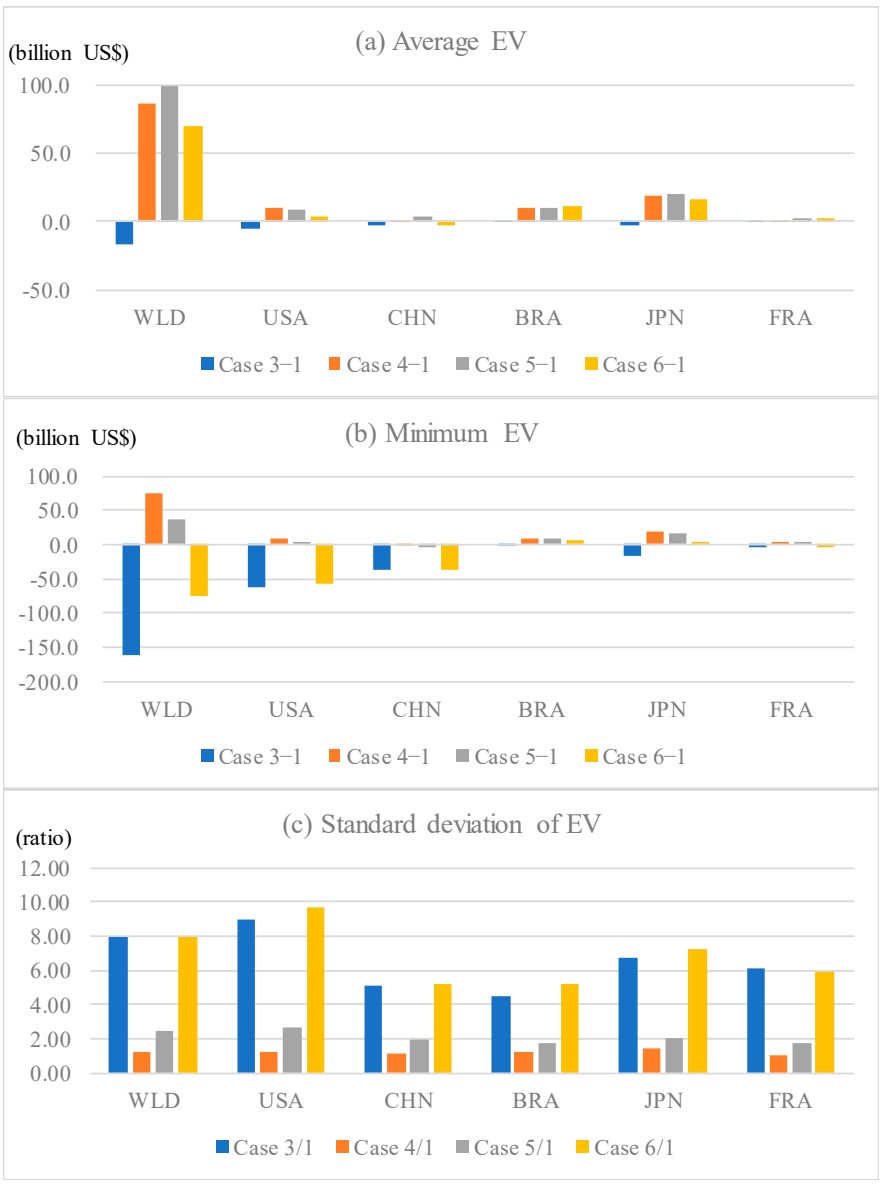

**Figure 5.** Changes in social welfare levels measured by equivalent variation (EV) due to SYC under future climate change and trade liberalization. Similar comparison on EV by regions and two crop models are shown in the Supplementary Materials (Figures S14 and S15).

## 4. Discussion and Policy Implications

Our simulation results show that SYCs among regions and crops were significant and robust and expanded the volatility of agricultural price and EV under future climate change. Hence, SYC becomes a cause of systemic risk to the global economy. Considering this risk under climate change, several policy implications can be noted.

When SYC becomes stronger due to future climate change, many countries will suffer from fluctuations in agricultural prices. Even food-exporting countries such as the United States will suffer from rising agricultural prices beyond the chronological trend and experience a decrease in the social welfare level in the event of an extreme year. An increase in the US agricultural prices would be led by an increase in agricultural exports, which are motivated by an increase in world agricultural prices during simultaneous global crop failure under future climate change. Hence, increasing global food stocks and developing high-temperature-tolerance varieties of food, noted in existing research, are evidently important, but realistically uncertain to achieve. In addition to these efforts, to reduce systemic risk, it is useful to secure a variety of agricultural production areas, both domestically and internationally. Countries such as Japan, where the food self-sufficiency rate is low, could benefit by maintaining a certain level of domestic food production capacity as well as keeping tight relations with many food-exporting countries.

Trade liberalization, as economic theory suggests, will generate profits and mitigate the negative effects of future climate change by increasing consumer surplus through agricultural price reductions and by increasing the income of exporting countries through trade. However, as measured in this study, the effects of trade liberalization could be reduced by SYC. Therefore, policy makers should consider that effects of trade liberalization would be overestimated if SYC is ignored.

Another policy that deserves consideration is enhancing the insurance system to compensate for global crop failures. When insurance plans are designed with consideration of systemic risk under climate change, premiums rise and become new costs for the global economy. For the insurance sector, a holistic framework to assess and mitigate systemic risk was proposed [35]. Similar considerations are required for systemic risk from climate change in the global food market. If the insurance system operates without considering systemic risk and its costs, the system itself is likely to fail when the risk is realized. Thus, private firms and policymakers must understand the risk and accordingly prepare in advance to manage synchronicity in the food system through a better understanding of SYC.

When policies consider systemic risk, the accuracy of climate, crop, and economic models that can predict the degree of risk is key. From an academic viewpoint, by modeling local extreme meteorological phenomena and local crop growth conditions, the causes of the SYC can be elucidated, which would help measures to avoid systemic risk. Increasing the accuracy of these models requires enormous costs. To improve the accuracy of crop models, e.g., we must develop crop-yield statistics with more detailed and localized units than statistics at the national level. Such statistics and models can be classified as the world's public goods, improving the social welfare level throughout the world. Therefore, it is important to promote research on model building in the field of crop yield and price prediction through international cooperation.

## 5. Summary and Conclusions

This study quantified the influence of systemic risk caused by SYC in the global food market under future climate change; it also evaluated the effects of trade liberalization when systemic risk exists, using a CGE model based on harvest predictions from crop models and global climate models. Simulation results demonstrate the following points.

First, the SYC structure was statistically robust among countries and four crops, and will be enhanced by climate change. Such global SYC is probably created by two common influences in regions that produce the same crop. The first is the rising or falling trend of crop yields due to the increase in the global temperature and $CO_2$ concentration, and the second is the widening of yield fluctuations as climatic conditions approach biological thresholds.

Second, where there was SYC under future climate change, agricultural price volatility increased more than would be expected in the random disturbance (non-SYC) case. In the United States, e.g., the highest price and the fluctuation range of agricultural price would be, respectively, 2.5 times higher and 10 times larger than in the non-SYC case. Hence, social welfare levels in most regions of the world are reduced by SYC.

Third, trade liberalization benefited both food-importing and exporting regions, because food-exporting regions increase domestic income due to an expansion of trade, whereas food-importing regions increase consumers' surplus due to a decrease in food price. However, such benefits of trade liberalization were degraded by SYC, causing unstable price fluctuations. For example, Japan, as one of the major importers, could decrease agricultural price by 10% after trade liberalization without SYC. However, these effects of liberalization would decrease to 8% by SYC in 2050–2099. Therefore, the effects of trade liberalization would be overestimated, if SYC is ignored.

These results demonstrate that SYC under climate change becomes a systemic risk for the global economy. Typically, influences of SYC are too small to be recognized in the market, but when significant change occurs, it leads to serious social welfare loss worldwide. Considering this risk, it is both prudent and important to review adaptation measures for climate change based on the quantitative results from economic and crop models as applied here.

There were limitations to this research. First, the GTAP data used were a 2011 version; an analysis with new data would be useful. Second, the analysis did not consider changes in capital stock or labor supply. When populations will increase in the future, the surge in agricultural prices should be further exacerbated due to global crop failure. An analysis considering population changes and dynamic analysis, where investment endogenously moves, would also be of interest. Finally, improving the accuracy of economic and crop models is important in and for the academic field.

**Supplementary Materials:** The following are available online at http://www.mdpi.com/2071-1050/12/24/10680/s1, Table S1: Frequency in magnitude of correlation coefficient (*r*) between regions by crop models. Figure S1: The average effective farmland productivity (EFP) of rice by regions and crop models. Figure S2: The average EFP of wheat by regions and crop models. Figure S3: The average EFP of other cereal grains including maize (GRO) by regions and crop models. Figure S4: The average EFP of oil seeds including soybean (OSD) by regions and crop models. Figure S5: The standard deviation of rice EFP by regions and crop models. Figure S6: The standard deviation of wheat EFP by regions and crop models. Figure S7: The standard deviation of GRO's EFP including maize by regions and crop models. Figure S8: The standard deviation of OSD's EFP including soybean by regions and crop models. Figure S9: Average agricultural price (P_agr) without trade liberalization comparing by regions and crop models. Figure S10: The standard deviation of agricultural price (P_agr) without trade liberalization comparing by regions and crop models. Figure S11: Average food net exports (exports–imports) by region (average of two crop models). Figure S12: Net effects of trade liberalization in average agricultural price (P_agr) comparing by regions and crop models. Figure S13: Net effects of trade liberalization in the standard deviation of agricultural price (P_agr) comparing by regions and crop models. Figure S14: Changes in average equivalent variation (EV) with and without synchronized yield change (SYC) or trade liberalization comparing by regions and two crop models. Figure S15: Changes in the standard deviation of EV with and without SYC or trade liberalization comparing by regions and two crop models.

**Author Contributions:** Conceptualization, Y.K.; methodology, Y.K. and G.S.; validation, Y.K.; formal analysis, Y.K., G.S. and T.I.; investigation, Y.K.; data curation, G.S. and T.I.; writing—original draft preparation, Y.K.; funding acquisition, Y.K. All authors have read and agreed to the published version of the manuscript.

**Funding:** This study was supported by the Grant-in-Aid for Scientific Research 16H04991, 16KT0036, and 20K06269 (Ministry of Education, Science, Sports and Culture).

**Acknowledgments:** The climate data for analysis were supplied by Motoki Nishimori (NARO), and English language editing was performed by Editage (www.editage.jp). The authors greatly appreciate their support.

**Conflicts of Interest:** The authors declare no conflict of interest.

## Appendix A

To eliminate the influences of technological progress, such as fertilizer effects and new variety creation, detrended yields ($\widetilde{YA}_{i,r,t}$) are calculated from the actual yield ($YA_{i,r,t}$) by $\widetilde{YA}_{i,r,t} = YA_{i,r,t}/(a_{i,r} + b_{i,r} \cdot t)$, where $i$, $r$, and $t$ represent the four crop categories, countries, and year, respectively. Here, "$a$"

and "*b*" are the intercept and slope, respectively, estimated from the regression between actual crop yields and the year trend, *t*.

**Appendix B**

EFP is assumed to change the farmland-input in the cost function related to the production function. The unit cost for input factors derived from cost minimization behavior of producers in Lanz and Rutherford [31] is modified as $c_{j,r}^f = \left( \sum_f \theta_f \cdot (p_{f,j,r}^{pf} / \gamma_{f,j,r})^{(1-\sigma)} \right)^{\frac{1}{(1-\sigma)}}$, where $c_{j,r}^f$ is the unit cost of factor, *f*, in sector *j* and region *r*; the suffix *f* shows input factors, i.e., labor (*lab*), capital (*cap*), land (*lnd*), and other resources (*res*); $\theta_f$ is the cost share of each input factor calibrated from the base year data; and $\sigma$ represents the substitution elasticity between input factors. Here, $p_{f,j,r}^{pf}$ is the factor price with taxes, and $\gamma_{f,j,r}$ is the input factor productivity in year *t*, as $\gamma_{f,j,r} = \begin{cases} 1, \ f \in lab, \ cap, \ res \\ EFP(j,r,t), \ f \in lnd \end{cases}$.

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
