# Peer review of "Systemic Risk in Global Agricultural Markets and Trade Liberalization under Climate Change: Synchronized Crop-Yield Change and Agricultural Price Volatility"

_sustainability, doi:10.3390/su122410680_

Round 1
Reviewer 1 Report
Study uses highly sofisticatet statistical modelling, however data used are not up to date,
Assumption that climate change will increase yield instability is appropriate, however why it will create "synchronised yield change" needs to be better enlighted. Some terms and definitions used in the study needs to be better explained.
Study has several limitations and shortcomings, eg. excluding European Union from the model which is the region of main area of world trade disturbances. There are needs of better describing expected or possible forms of trade liberalisation or intervention in different regions.
Resewarch results are interesting , logoc and justified. Results has higher academi than practical value.
Author Response
To Reviewer 1,
Thank you very much for your sincere comments and suggestions. According to your comments, we revised our paper. However, we could not follow some of your comments. Our reply and treatments are as attached file. We are very pleased, if you can understand our reply. Thank you again. Best regards.

Reviewer 2 Report
The papers is one of the high scientific soundness and analytical accuracy. The authors properly explain what they have discovered in the research. Results are clearly laid out in a logical manner.
Some suggestions to improve the paper:
The analysis of systemic risks in global food markets under future climate change requires the assessment of both probability and impact of the observed phenomenon. The paper addresses in a very comprehensive and multifaceted approach the likely impact of synchronized crop-yield change on agricultural price volatility and trade liberalization benefits, but has easily overlooked the probability component. At the same time, adding into the analysis the degree of vulnerability and the rate-of- rise of the process can provide relevant information for the full picture over the systematic risk.
The paper could improve its hypothesis through references of analyses that have quantitatively assessed if global crop production has actually tended towards synchronized failure. For instance, Mehrabi and Ramankutty (2019) show that synchronization in production within major commodities such as maize and soybean has declined in recent decades, leading to increased global stability in production of these crops. Other papers suggest that historical records show that for major commodities such as maize and soy, global crop production systems have not tended towards synchronized failure.
The policy implications section could better address the need for a better understanding of how to engineer or maintain asynchrony into the food system wherever possible. At the same time, there are locations in the world, which are stabilizing global food production, and there is also some empirical evidence that the food system has become less synchronized over time.
Line 130. How was the year 2000 determined as the proper year to eliminate the effects of the technological progress and according to which methodology.
Line 361-363. The net export increased in most food exporting regions...after trade liberalization. This sentence should be sustained by previous studies or reports.
In section summary and conclusions, the authors should compare their results with the results in the previous studies and to highlight the results that sustained or not previous s results.
Author Response
To Reviewer 2,
Thank you very much for your sincere comments and suggestions. According to your comments, we revised our paper. However, we could not follow some of your comments. Our reply and treatments are as attached file. We are very pleased, if you can understand our reply. Thank you again. Best regards.

Reviewer 3 Report
This paper investigates two very important issues. The first is whether climate change is likely to increase the risk of synchronised yield changes across countries over time (using four major crops for investigation) thus become a case of ‘systemic risk’. The second is whether and how such increased risk is likely to affect the stabilising role of international trade.
The methodological approach seems appropriate. Weather impacts of climate change in three periods (T1 2007-2014, T2 2015-2050 and T3 2051-2100) are downloaded from five different climate models according to the RCP8.5 pathway. These weather impacts are then input into two different crop models, also considering the CO2 fertilisation effect, to generate annual crop-yield forecasts for each region from 2007 to 2099. The deviation in the annual country-crop yields from the reference period (taken as the average country-crop yields in the T1 period 2007-2014) is then calculated for each year in periods T2 and T3. The procedure for T1 is different as the intention is to generate random and regionally independent shocks to replicate a non-SYC scenario. The explanation (lines 190-196) was not clear enough for me to understand the procedure.
These deviations are then applied as shocks to effective farmland productivity in a global CGE model to generate price and economic welfare deviations from the baseline. Furthermore, the CGE model simulations are run in two variants (a) with existing trade barriers in place (b) a trade liberalisation scenario in which all tariffs and export subsidies on agricultural commodities are removed.
Unfortunately, I am not able to assess the results in this paper as I was not able to follow the explanation of the methodology. This may be partly due to the fact that the English language used in the paper is difficult, but I also think it is due to incomplete explanation of key steps.
Having described the climate models and the crop models on Page 3, the key paragraph to explain the methodology is lines 139-144. This paragraph begins “The analysis.. calculated the correlation coefficients of yields in T1, T2 and T3 periods.” I assume this should refer to the deviations in yields rather than absolute levels of yields, but how the deviations are calculated for period T1 is not explained. Furthermore, the sources of yield variation (and thus deviations) is not explained. It is not clear if the crop models, once populated with the relevant weather information and CO2 levels, give a discrete yield for each country-crop-year combination. If so, as the input data for the crop models only changes very slowly over time, although accelerating with rising temperature towards the end of the period, there should be very little change in the spatial correlation within the T2 and T3 periods. The correlation coefficients calculated should be defined more carefully.
In the next paragraph (lines 145-148) the focus moves to the degree of SYC robustness or “chronological robustness” (line 94). This concept is also not defined, or what its significance is. The paper says that “we selected statistically significant combinations of regions and/or crops” but there is no explanation of what a statistically significant combination is or how it is defined. If we have two regions A and B, it seems we have different estimates of yield (or yield deviation) for each crop and each year depending on the climate and crop model used to generate it. I was not clear if this was the basis for determining statistically significant combinations of regions and/or crops and, if so, why we should be interested in this.
Similarly, the results are presented in terms of the frequency of the correlation coefficients between regions (line 223), without it being clear which variables were being examined. As a result, I am not able to comment on the results.
Editorial issues:
- The authors refer to the impact of climate on agricultural production but it is not clear if they mean weather. Climate is usually defined as 30-year averages and annual data are referred to as weather. Climate change of course affects weather conditions but climate shocks (line 80) are better described as weather shocks.
- Line 72. In discussing portfolio theory, the authors state that “a crash in one brand in the basket induces a crash in another brand, a domino effect”. Not necessarily. In the case of yield variations, it is not that a poor harvest in Argentina induces a poor harvest in France, but that there may be a correlation, not necessarily a causation.
- Line 116. The basis for the reproducibility of this crop model’s estimates is not explained. Are these figures calculated on the basis of country-years, or country averages, or global averages? The reference on line 120 to FAO’s statistical data on a global scale suggests it may be production-weighted global averages but I could not tell.
- Line 154. In the aggregated regions, there is reference to ‘Rest of EFTA’. I suspect this may be ‘Rest of EU’ as none of the countries mentioned previously are members of EFTA.
- Line 183. I am assuming that it is a static GTAP model with base 2011 that is kept constant throughout, this should be made explicit.
- Line 233. I was surprised by the suggestion that yields of the same crop will simultaneously increase or decrease. My understanding is that this will depend on initial conditions. In higher latitudes with lower temperatures, warner temperatures may increase yields but decrease them in lower latitudes.
- Line 300. Here we are told first that the data from the two models are pooled, but in the next line that the authors were careful not to mix data from different crop models. How can we interpret this apparent inconsistency?
- Line 395-395. My advice would be to avoid the use of normative descriptors (positive, negative) here and use more neutral terms (upward, downward) as whether a price movement is positive or negative is very much in the eye of the beholder (producers and consumers are likely to take opposite views).
- Line 451. Statements are made here that mix up trends in price levels (not the focus of this paper) and fluctuations in prices.
- Line 545, 552. Subscripts, superscripts and variable names seemed to be missing here.
I regret that I cannot give a more qualified assessment of the results. My recommendations for improvement would focus on (a) giving a much clearer explanation of the basic methodology, defining the various concepts used and how they are applied in the analysis, and (b) seeking to improve the clarify of the English language.
Author Response
To Reviewer 3,
Thank you very much for your sincere comments and suggestions. According to your comments, we revised our paper. However, we could not follow some of your comments. Our reply and treatments are as attached file. We are very pleased, if you can understand our reply. Thank you again. Best regards.
